# Salespeople vs SalesBot: Exploring the Role of Educational Value in Conversational Recommender Systems

**Lidiya Murakhovs'ka**    **Philippe Laban**    **Tian Xie**
**Caiming Xiong**    **Chien-Sheng Wu**
Salesforce AI Research
{l.murakhovska, plaban, t.xie, cxiong, wu.jason}@salesforce.com

## Abstract

Making big purchases requires consumers to research or consult a salesperson to gain domain expertise. However, existing conversational recommender systems (CRS) often overlook users' lack of background knowledge, focusing solely on gathering preferences. In this work, we define a new problem space for conversational agents that aim to provide both product recommendations and educational value through mixed-type mixed-initiative dialog. We introduce SalesOps, a framework that facilitates the simulation and evaluation of such systems by leveraging recent advancements in large language models (LLMs). We build SalesBot and ShopperBot, a pair of LLM-powered agents that can simulate either side of the framework. A comprehensive human study compares SalesBot against professional salespeople, revealing that although SalesBot approaches professional performance in terms of fluency and informativeness, it lags behind in recommendation quality. We emphasize the distinct limitations both face in providing truthful information, highlighting the challenges of ensuring faithfulness in the CRS context. We release our code and make all data available [1].

## 1   Introduction

Conversational recommender systems (CRS) use multi-turn dialog to understand user preferences, gather information, and provide suitable recommendations (Gao et al., 2021). They have gained significant attention from academia and industry due to their flexibility compared to one-shot recommender systems. CRS allow users to dynamically refine their preferences and express feedback on recommended items through chat.

While many traditional CRS are built under a "System Ask-User Answer" paradigm (Zhang et al., 2018b), some recent works have identified the critical need to support mixed-type, mixed-initiative

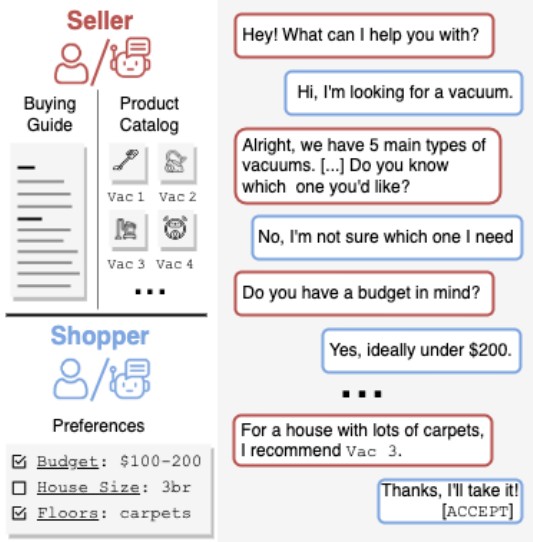

Figure 1: The SalesOps framework is a conversational recommendation system simulation involving: (1) a Seller viewing a buying guide and product catalog, (2) a Shopper gradually learning about shopping preferences. Either can be simulated using an LLM-based system.

dialog and assist users with underspecified goals (Kim et al., 2020; Liu et al., 2020, 2022). However, most CRS remain focused on domains involving content recommendation such as movies, books, music, news, etc, which require a different type of recommendation strategy than the e-commerce space where consumers often need a certain level of background knowledge to understand their options (Laban et al., 2020; Papenmeier et al., 2022). In content recommendation domains, CRS can achieve success by questioning a user about previous content consumption and retrieving similar content. This strategy is not valid for the sale of complex products, as prior user habits do not inform a buyer's item-specific preferences.

In this work, we focus on complex products with multiple attributes that would usually require significant expertise or a salesperson consultation to make an informed purchase decision on - e.g., TVs,

---

[1] https://github.com/salesforce/salesbot

guitars, etc. We associate the ambiguous user goals often present in this real-life setting with the lack of knowledge about the product domain and suggest the need for the agent to provide educational value in addition to fulfilling the recommendation objective. We address this new problem space by introducing SalesOps, a framework that facilitates the recreation of realistic CRS scenarios and agent evaluation, as shown in Figure 1. Our method incorporates several novel features.

- We utilize existing buying guides as a knowledge source for the sales agent. We provide the seller with a relevant guide in addition to a product catalog, which enables them to educate shoppers on the complex product space.

- Shopping preferences are *gradually* revealed to the shopper during the course of conversation in order to simulate the underspecified goal scenario of a typical uninformed shopper.

- A multi-dimensional evaluation framework is designed to measure sales agent performance in terms of (a) quality of final recommendation, (b) educational value to the shopper, and (c) fluency and professionalism.

We leverage recent progress made in large language models (LLMs), which has enabled increasingly powerful conversational agents, to build a pair of agents - SalesBot and ShopperBot which can simulate either side in the SalesOps framework. Thus, facilitating evaluation in any of the following settings: human-human, human-bot, bot-human, and bot-bot. We critically examine the components of these agents to understand where modern LLMs have the most impact on CRS.

We recruit 15 professional salespeople to study the gap between human experts and SalesBot at complex product conversational recommendations. The results reveal that although SalesBot matches professionals in tone politeness and educational quality, there remain minor gaps in the quality of recommended products. We also perform a preliminary analysis of faithfulness within the SalesOps framework, which is important in domains involving generative AI as they enter the applied setting. Our NLI-based analysis (Laban et al., 2022) reveals that salespeople can use strategies that may seem unfaithful, to upsell or simplify technical details. These results highlight the practical challenges of implementing robust faithfulness checks in domains like conversational sales.

## 2 Related Work

### 2.1 Conversational Recommendation

Since the introduction of conversational recommendation systems (Sun and Zhang, 2018), the field has grown to a wide range of task formulations, settings, and application scenarios (Gao et al., 2021; Deng et al., 2023). Recent works explore conversational systems in the context of multi-type and mixed-initiative dialogs (Liu et al., 2020, 2022). They release sizable datasets in Chinese to advance the field toward supporting users with undefined goals. However, these primarily cover content recommendation domains such as movies, food, and news. As such, they do not directly address the challenges the e-commerce domain poses. In this work we extend these ideas to the specific context of e-commerce, where the focus is on providing recommendations for complex products.

In the e-commerce space, Fu et al. (2020) presented COOKIE - a CRS dataset constructed synthetically from user reviews. Most recently, Bernard and Balog (2023) presented a small multi-goal human-to-human conversational dataset with 64 chats for e-commerce. While closely related to our task, they do not aim to simulate ambiguous user preferences or target the educational objective; instead, they reveal all user requirements at once.

While existing CRS are effective, criticism can be raised regarding the incorporation of expert knowledge into the user experience. While some systems provide explanations for recommendations (Chen et al., 2020; Wen et al., 2022), their primary focus is not on educating the user about the different options available. This highlights the need for research into new methods of integrating expert knowledge into CRS.

### 2.2 Knowledge-Grounded Dialog

Knowledge-grounded response generation in dialogue systems has been explored for years in both task-oriented (Madotto et al., 2018; Li et al., 2022) and open-domain systems (Zhang et al., 2018a; Shuster et al., 2022a; Thoppilan et al., 2022; Shuster et al., 2022b). However, it is important to note that most existing approaches focus on passively responding to user queries rather than proactively conveying the knowledge.

A study by Cai et al. (2022) proposes a teacher-bot that transmits information in passages to students, aiming to help them acquire new knowledge through conversation. While their work focuses

solely on the educational objective, our work combines it with CRS, suggesting suitable products while educating users about the relevant domain.

Some CRS have incorporated knowledge graphs (KGs) as complementary resources for analyzing item- and attribute-level user preferences (Zhang et al., 2021; Zhou et al., 2021; Ren et al., 2023). The primary objective of KGs is to enhance recommendations or aid preference elicitation rather than assisting sellers in answering inquiries or proactively conveying knowledge. Other works (Schick et al., 2023; Peng et al., 2023) focus on augmenting LLMs with external knowledge to address common pitfalls like hallucinations and lack of up-to-date information. This can greatly benefit our goal; thus, we leverage and augment LLMs with external tools to build conversational agents in this work.

## 2.3 Evaluation and User Simulation

Designing effective evaluation protocols has long been a challenge for various conversational systems (Deriu et al., 2021). Interacting with real users is costly, prompting the adoption of user simulators to assess proactive interactions in dialogue systems (Deng et al., 2023). Sekulic et al. (2022) utilized a GPT-2-based generative agent to simulate users in Conversational Question Answering (ConvQA) setting. We build upon their approach and incorporate the latest advancements in LLMs. We also introduce a novel concept of gradually incorporating user preferences into the simulation process.

## 3 SalesOps Framework

We now describe the SalesOps framework, allowing us to study CRS systems in terms of educational and recommendation objectives for complex product scenarios. In SalesOps, two actors – the Seller and the Shopper – have a conversation that begins with a Shopper request and ends once the Seller makes a product recommendation that the Shopper accepts. Each actor gets access to specific content elements that assist them in completing the task. We first describe the three content elements and the procedure used to generate them at scale, and then we describe the constraint we put on actors' access to the content to create realistic sales conversations.

## 3.1 Content Elements

As illustrated in Figure 1, for any given product (i.e., vacuums), three content elements are required to instantiate the SalesOps framework. The Product Catalog and the Buying Guide are the content elements accessible to the Seller, and the Shopping Preferences are accessible to the Shopper.

Importantly, we populated all content elements for the six product categories we include in our initial release, but we aim for the procedures to be automatable, so they can be expanded to new products, unlike previous resources that can become outdated (Ni et al., 2019).

### 3.1.1 Product Catalog

In the SalesOps framework, the Seller has access to a fixed list of products that can be recommended to the Shopper. Each product consists of (1) a unique ID, (2) a product name, (3) a price, (4) a product description, and (5) a feature set.

When creating the product catalogs, we initially considered leveraging the Amazon Product Reviews dataset (Ni et al., 2019). However, we found that the products in the dataset are outdated (1997-2018), which greatly impacts the conversations we obtain from a study with human participants in 2023. In many cases, product information and pricing are obsolete, products with the latest technology are missing (e.g., QLED TVs) and thus, the product catalog misaligns with updated Buying Guides and participant expectations.

We generate synthetic product entries using an LLM since web-scraping an up-to-date product catalog can lead to limitations in terms of open sourcing. We first repeatedly prompt the LLM to generate a diverse list of an average of 30 product names for a given category (e.g., TVs). We then prompt the model for each product name to generate realistic product metadata, including a title, description, price, and feature list. Appendix A presents more details of this process.

Unlike previous approaches that utilize databases with thousands of items, we deliberately limit the product catalog to approximately 30 items per category to mirror the curated selection of a typical store (Fu et al., 2020; Bernard and Balog, 2023). This also significantly impacts human sellers' ability to complete the task as they need to familiarize themselves with the products available to effectively perform their role. However, the automated nature of the creation process would allow us to expand the product catalog efficiently.

During a SalesOps conversation, the Seller can decide on their turn to recommend one or several items whose details will be included in a subsequent message to the shopper (see Appendix C for

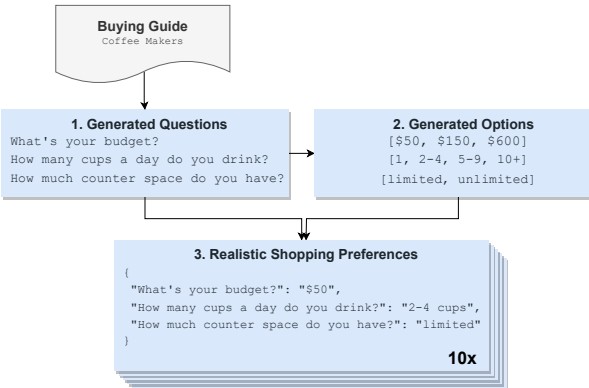

Figure 2: Shopper Preference Generation Pipeline using ChatGPT. Example for the Coffee Makers category.

an example conversation).

### 3.1.2 Buying Guide

Professional salespeople often receive training or rely on technical documentation to effectively sell complex products. We proxy this expert knowledge through leveraging publicly available buying guides. Buying guides, such as ones available on BestBuy[2] or Consumer Reports[3], are often written by professionals to help coach buyers on the decision-making process so that they can determine the best option for themselves. For each product category, we retrieve the top five articles from the C4 corpus (Raffel et al., 2019) that match the search query "[PRODUCT] Buying Guide", and select the most appropriate one.

On average, the buying guide we select for each product category is 2,500 words and 50 paragraphs long. Selected buying guides are diverse in their organization, with some being organized by shopper persona, shopping budget, or major item subcategories (e.g., drip vs. espresso machines). The heterogeneity in the layout of buying guides goes towards creating a realistic experimental setting in which the layout of knowledge documents for a particular product might not be known in advance.

### 3.1.3 Shopping Preferences

Figure 2 introduces the three-step method we use to obtain shopping preferences for a given product category, which relies on LLMs. First, we generate a list of five possible questions the Seller may ask based on the buying guide (e.g., "How many cups of coffee do you drink per day?"). Second, we generate for each question a set of answer options

(e.g. ["1", "2-4", "5-9", "10+"]). Although we attempt to have mutually exclusive questions, it is inevitable for some combinations to be improbable (e.g., a very high-capacity coffee maker for the smallest budget), thus, we leverage LLMs in a third step to select 10 diverse but realistic combinations of the preferences.

Unlike prior work that reveals the shopper preferences in their entirety when the conversation is initiated (Liu et al., 2022; Bernard and Balog, 2023), we choose to reveal preferences to the Shopper gradually during the conversation, providing a more realistic simulation of an underspecified buying experience (Kim et al., 2020).

To achieve this objective, for each Shopper turn in the conversation, we extract the last Seller message, and use a semantic similarity model[4] to detect whether the utterance corresponds to a question related to one of the preferences. If the similarity passes a manually selected threshold, the related preference is revealed to the Shopper, and they can choose to leverage the additional information in their response. We note that the system reveals at most one preference per Shopper turn and does not enforce that all preferences are revealed. We intend these choices to simulate a realistic conversational experience for the Shopper and Seller.

### 3.2 SalesOps Actors

In SalesOps, the two actors – the Seller and the Shopper – can either be simulated by an LLM-based system or enacted by a person such as a sales professional or crowd worker. We briefly introduce the considerations for each actor.

The **Seller** has access to the Product Catalog and Buying Guide during a SalesOps conversation, corresponding roughly to 65 paragraphs which is a large amount of content both for humans and a system enacting the role. We estimate that a human enacting this role would require roughly 30 minutes at an average reading speed of 200 words per minute to read all the content. In our experiments with professional salespeople, they were each provided a period of reading time to get familiar with the content prior to participating in conversations. In our automated Seller implementation, we leverage a retrieval system to efficiently provide the content to the LLM components with limited context lengths.

---

[2] https://bestbuy.com/
[3] https://www.consumerreports.org/

[4] sentence-transformers/ paraphrase-multilingual-mpnet-base-v2

Appendix D goes over the SalesOps Seller User Interface, built using the Mephisto library (Urbanek and Ringshia, 2023). The interface is extended for several aspects: (1) contains the Buying Guide and a search interface to the Product Catalog, (2) the Seller must select which paragraphs of the Buying Guide they are leveraging in crafting their response (if any), (3) the post-chat questionnaire which asks to select utterances where Shopper revealed their preferences and rate the conversation partner.

During a chat, we further log all product search queries. As a result, the metadata generated when implementing the SalesOps framework can be useful for tasks beyond the CRS setting, such as knowledge-grounded response generation, conversational summarization, and query generation.

On the other hand, the **Shopper** is only provided with the product category they are tasked with shopping for at the initial stage of a SalesOps chat. Shopping preferences are revealed based on the Seller's questioning as the conversation unfolds. The Shopper interface, presented in Appendix D requires fewer adaptations of Mephisto than the Seller interface: (1) we repurpose one-sided "Coordinator" messages to reveal Shopping Preferences during the conversation, and (2) when a product recommendation is suggested by the Seller, the Shopper interface displays buttons to accept or reject the item.

## 4 Bots Implementation

We now present ShopperBot and SalesBot, the LLM-based implementations (ChatGPT in our experiment) that can simulate both sides of the SalesOps framework.

### 4.1 ShopperBot

ShopperBot's goal is to generate responses in accordance with the provided set of preferences ($P$), consisting of several question-answer pairs ($q$-$a$). We achieve this objective by prompting LLMs with (a) natural language instruction to act as a shopper seeking [PRODUCT] (e.g., a TV), (b) a list of currently revealed shopping preferences, and (c) the chat history, at every turn in the conversation. Full prompt can be seen in Appendix B.

When the latest seller's utterance includes a recommendation of an item, ShopperBot is instructed to include [ACCEPT] or [REJECT] token in its reply. It will base this decision on the whole set of preferences ($P$) to ensure consistency with the simulated

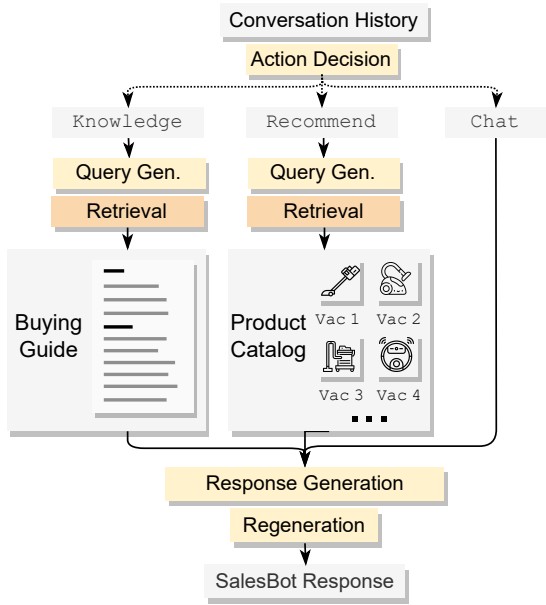

Figure 3: SalesBot architecture to generate a Seller response, based on two Content Elements and the conversation history. LLM-based modules are in yellow.

scenario - i.e., if too few preferences were revealed at this point in the conversation, we do not want the shopper to accept an item that would not satisfy the whole set $P$.

We note that the Shopping Preferences are far from a comprehensive listing of all questions that could occur when shopping for a complex product category. The ShopperBot is instructed to make its own decisions when choices are not in $P$ (e.g., the preferred color of a coffee machine) and fluently converse with the Seller.

Qualitatively, we have confirmed in our experiments that ShopperBot can provide subjective and unique preferences, realistically simulating a human shopper.

### 4.2 SalesBot

SalesBot has access to two main external tools - 1) knowledge search and 2) product search. As shown in Figure 3, it generates dialogue responses using a series of modules. Each module is an independent component of the overall system.

#### 4.2.1 Action Decision

This module decides which tool to use based on the current conversation history. Selecting from the following: *Knowledge Search* (details in section 4.2.2), *Product Search* (details in section 4.2.3), *Response Generation* (details in section 4.2.4).

We query an LLM to make this choice and pro-

vide natural language instructions on when to use each of the available tools in our prompt.

### 4.2.2 Knowledge Search

*Knowledge Search* module's main purpose is to educate the user by incorporating expert domain knowledge into the conversation. It is made up of two components: 1) query generation, and 2) retrieval. We ask the LLM to generate a query based on the chat history.

We then use a FAISS retriever to lookup relevant knowledge article paragraphs (Johnson et al., 2019). We concatenate top 3 paragraphs (separated by "\n\n") and feed it as external knowledge to the *Response Generation* module described in section 4.2.4.

### 4.2.3 Product Search

*Product Search* module's goal is finding relevant items to recommend to the user. Similar to the above, this module is made up of 1) query generation, and 2) retrieval. We embed each product's information (i.e., title, description, price, and feature list) using the Sentence Transformer model [5] and retrieve the top 4 products based on the query embedding (obtained using the same model). Same as in *Knowledge Search*, we concatenate the results. Note that in some cases, we may not need all 4 products, for example, if the shopper asked a follow-up question about a certain product, the response should not include all 4 retrieved items. We leave it up to the *Response Generation* module to determine which products should be mentioned based on the chat history.

### 4.2.4 Response Generation

Based on the Action Decision module, response generation can either include external information (e.g., buying guide excerpts, product information) or not. We thus, write two separate prompts to respond to the shopper. Response Generation *with External Knowledge* is based on: Chat History, Action Selected, Query Generated, Retrieved Results. Response Generation *without External Knowledge* is based solely on the chat history.

We additionally implement a *Regeneration* submodule to rewrite the final response if needed. We place a limit on `max_tokens_generated` when prompting the LLM and ask it to rewrite the previously generated response if it was cut off due

to length. This forces SalesBot's responses to be concise and contain full sentences.

## 5 Evaluation Criteria

Along with the SalesOps framework, we propose a multi-dimensional evaluation that defines success for the Seller along three axes: (1) **recommendation quality**, which verifies whether the recommendations of the Seller are compatible with the Shopper's preferences, (2) **informativeness**, which checks whether the Seller provides educational value to the Shopper, and (3) **fluency** which evaluates the Seller's ability to communicate professionally and concisely. We next define the metrics – both automatic and manual – used to evaluate each axis.

Recent work shows the promise of using LLMs for evaluation across many NLP tasks (Liu et al., 2023; Fu et al., 2023; Laban et al., 2023). Following this thread of work, we leverage GPT-4 (OpenAI, 2023) for automatic evaluation purposes in several of our proposed metrics.

### 5.1 Recommendation Quality

Accurate recommendations that match shopper preferences are a core expectation of CRS. The authors of the paper manually annotated each product category and its 10 corresponding Shopping Preferences for all acceptable product recommendations as the ground truth. On average, a given shopping preference configuration yielded 4 acceptable products from a product catalog of 30 items. Thus, for a completed SalesOps conversation, we compute recommendation accuracy ($Rec$).

### 5.2 Informativeness

We propose two metrics to measure the informativeness of the Seller during a conversation.

First, we leverage an NLI-based model to measure the content overlap between the Seller's utterances and the buying guide, as such model has been shown to perform competitively on tasks involving factual similarity (Laban et al., 2022; Fabbri et al., 2022). Specifically, we calculate the % of the buying guide sentences that are entailed at least once by a seller utterance ($Inf_e$).

Second, we assess the shopper's knowledge through a quiz that we designed which consists of 3 multiple-choice questions that can be answered using the buying guide (examples in Appendix C. We then ask crowd-workers to an-

---

[5]`sentence-transformers/all-mpnet-base-v2`

| Agent | $Flu_e$ ↑ | $Flu_i$ ↑ | $Inf_e$ ↑ | Rec ↑ |
|---|---|---|---|---|
| Ada-RG | 1.41 | 0.00 | 0.16 | 0.16 |
| Rule-AD | 4.82 | 0.59 | **0.86** | 0.40 |
| Key-QG | 4.97 | 0.72 | 0.71 | 0.36 |
| No-ReGen | **4.99** | 0.85 | 0.74 | **0.47** |
| SalesBot | **4.99** | **0.91** | 0.74 | 0.44 |

Table 1: Results of the ablation study of SalesBot components using GPT4-based evaluation metrics.

swer each knowledge question solely considering the conversation, with the option of choosing "Cannot answer based on the conversation". We report the % of correct answers on the knowledge questions ($Inf_q$).

## 5.3 Fluency

We frame two questions to measure the fluency and professionalism of the Seller:

$Flu_e$: How would you rate the salesperson's communication skills? (scale: 1-5)

$Flu_i$: Do you think the seller in the given chat is: (i) human or (ii) a bot? (Yes/No)

We perform annotation for the two fluency metrics both manually by recruiting crowd-workers (see Appendix C) as well as by prompting GPT-4 to answer both questions.

## 6 Ablation Study

SalesBot contains 4 LLM-based components, which we swap with baselines to understand the impact of using an LLM for each component.

### 6.1 Baselines

We implement four ablations of SalesBot:

**Ada-RG** We replace ChatGPT with the smaller GPT3 `text-ada-001` model in the response generation module.

**Rule-AD** We replace the Action Decision module with a rule-based system: select the "Knowledge" action for the first 6 turns, and the "Recommend" action afterwards.

**Key-QG** We replace Query Generation modules with a keyword method: extract five keywords from the latest utterance and concatenate them as a query.

**No-ReGen** We experiment with removing Regeneration module from response generation.

## 6.2 Results

We generated 150 conversations on the six product categories between ShopperBot and the five versions of SalesBots, and computed automatic results which are summarized in Table 1. Overall, **the use of an LLM is most crucial in the Response Generation component but is beneficial across all components.**

Replacing ChatGPT with a smaller LLM in the Response Generation component leads to the largest degradation across the board, confirming that smaller models are unable to handle the complex task of tying together the conversation history and external knowledge to generate a concise, coherent, fluent, and informative response.

The rule-based action decision module leads to improved informativeness (as the system relies more heavily on the knowledge action), at the cost of fluency and recommendation quality.

The keyword-based retrieval ablation leads to lower a 20% decrease in recommendation quality, confirming that generation of retrieval queries benefits from the generative flexibility of an LLM. Finally, adding the regeneration component leads to a boost in fluency, at the cost of a minor recommendation quality drop.

As a result, we recommend that designers of conversational agents: (1) Leverage LLMs in Response Generation, (2) Integrate Generative Flexibility in Retrieval Queries, and (3) Utilize Regeneration for Improved Fluency.

## 7 Salespeople vs SalesBot

We aim to study the qualitative differences between SalesBot and professional salespeople to comprehend the effect of deploying such systems in real-world settings (Guo et al., 2023), and perform an extensive human evaluation of SalesBot and Salespeople within the SalesOps framework.

### 7.1 Experiment Setup

We recruit 15 professional salespeople across a diverse set of industries (e.g., insurance, retail, B2B sales) through UserInterviews [6]. They were given a 1-hour onboarding session covering the SalesOps framework, reading a Buying Guide and Product Catalog, and completing warmup conversations. Participants then completed up to 15 SalesOps conversations with ShopperBot, which took an average of 3 hours.

---

[6]https://www.userinterviews.com/

| Statistics (avg) | 🖥️ | 🧑 |
|---|---|---|
| Nb. words of Seller utterance | 62.5 | 35.3 |
| Nb. words of Shopper utterance | 20.8 | 20.7 |
| Nb. of turns | 11.9 | 12.9 |
| Nb. turns before the first rec. | 6.0 | 7.9 |
| Nb. of recommendations | 2.6 | 2.4 |
| Nb. of triggered revelations | 1.7 | 2.8 |
| % Correct recommendations ($Rec$) | 44 | **54** |
| Information Quiz Score ($Inf_q$) | **32.9** | 31.8 |
| Fluency Score ($Flu_e$) | **4.4** | 4.2 |
| % Is Human ($Flu_i$) | 55 | **80** |

Table 2: Comparison of 🖥️ SalesBot and 🧑 Salesperson on conversation statistics (top) and human evaluation metrics (bottom).

In parallel, we generated 150 SalesOps conversations between SalesBot and ShopperBot for the same set of preferences. Unlike the ablation study in which we perform the evaluation with GPT-4, we recruit crowd workers from Amazon MTurk [7] to complete evaluation (interface in Appendix C).

## 7.2 Results

Table 2 presents statistics and evaluation results of the comparison between professional Salespeople and SalesBot. Overall, SalesBot's utterances are almost twice as long. It makes its first recommendation earlier and makes slightly more recommendations in total than professional salespeople.

Looking at the human evaluation, crowd workers were largely able to distinguish between Sales-Bot and professionals, as they were much more likely to believe the Seller was human for professionals (80%) than for SalesBot (55%), yet Sales-Bot achieved a higher Likert fluency score. This is likely due to **salespeople acting more casual in conversations**, making occasional typos which come across as less professional.

Finally, even though professionals write utterances that are nearly half the length, they achieve higher recommendation quality and almost equal informativeness. This pair of results confirms that **there is still a large gap between LLMs and professional salespeople with respect to conciseness.**

## 7.3 Faithfulness

In absolute terms, both SalesBot and Salespeople achieve low rates of correct recommendations (< 55%). We perform a qualitative analysis of failure

---

**GPT-4 Explanation of Unfaithful Content**

**Salesperson.** The salesperson incorrectly mentioned that the Cuisinart DCC-3200P1 has a hot water dispenser, which is not mentioned in the document.

**SalesBot.** The salesperson incorrectly stated that some drip coffee makers and espresso machines have the option for pre-measured pods or capsules. The document only mentions single-serve coffee makers as having the option for pre-measured pods.

Table 3: Example of GPT4-based detection of unfaithful behavior in both 🖥️ SalesBot and 🧑 Salespeople.

cases and find that there are several reasons for mismatches, including salespeople upselling products (i.e., convincing Shoppers to accept a product beyond their initial budget). We study this phenomenon through the lens of faithfulness analysis.

We provide GPT-4 with Buying Guides, the Product Catalog and the full conversation, and prompt it with identifying whether the Seller provides advice that is inconsistent with existing information. It detects that for both types of Sellers, roughly one in four chats contains unfaithful claims.

We provide examples of identified inconsistencies in Table 3. Our findings suggest that while SalesBot may occasionally hallucinate due to known LLM-bound limitations, **salespeople can also exhibit unfaithful behavior**, usually in the form of (a) upselling and (b) answering follow-up questions without knowing the true answer. Both are motivated by the desire to close the sale; however, this confirms the challenge of evaluating faithfulness in the sales domain, for which successful sales strategies might require unbacked claims.

## 8 Conclusion

In this paper we introduce SalesOps, a flexible framework for simulating realistic CRS scenarios in the context of complex product sales, and propose an evaluation paradigm for such systems. Our framework provides researchers and practitioners with a valuable tool for assessing the effectiveness and performance of conversational agents in sales settings. By developing SalesBot and ShopperBot within this framework, we gain insights into the individual components and their impact on conversational performance. Through a comprehensive

---

[7]https://www.mturk.com/

human study, we identify gaps in product recommendation quality and provide a faithfulness analysis of both automated and human sellers. These contributions advance the understanding and development of CRS, paving the way for improved sales interactions and user experiences.

## 9 Limitations

In this work, we heavily on LLMs to build the conversational agents (SalesBot and ShopperBot) within our framework. While LLMs have shown significant advancements in generating human-like responses and engaging in multi-turn conversations, they still suffer from hallucinations as shown in our faithfulness analysis, and thus impact the overall performance of the system.

Additionally, our evaluation primarily focuses on three aspects: the quality of the final recommendation, educational value to the shopper, and fluency/professionalism. While these aspects are important, there are other dimensions that could be relevant, such as user satisfaction, persuasion skills, and diversity of recommendations. We leave further exploration into CRS evaluation to future work.

We limit our evaluation to the chat-based sales experience, even though most sales conversations involving a shopper and a salesperson happen in audio form, either in a physical store or on the phone. Prior work has shown that adapting conversational content to the audio format is non-trivial (Kang and Gretzel, 2012), and requires modifications to remain natural and engaging (Laban et al., 2022). We leave it to future work to further adapt SalesOps and SalesBot to the audio setting.

## Ethical Statement

The ethical considerations in this work primarily revolve around the interactions with human participants and the potential implications of deploying conversational agents in real-life settings.

The study involved 15 professional salespeople who were recruited through User Interviews platform and participated voluntarily. The study aimed to ensure representation and inclusivity in its participant selection process. We recruited individuals of different genders, professionals of all levels (associate, manager, director, VP), and spanning a wide range of age groups from 19 to 65. They received compensation for their time and effort. An onboarding session was conducted to explain the task instructions and provide sample chats. Afterward, the participants had the freedom to complete the study at their own pace. Additionally, human evaluation was conducted with Amazon Mechanical Turk workers, who were compensated for their contributions.

We discuss the potential impact of deploying conversational agents, such as SalesBot, in real-life scenarios. The study highlights both the strengths and limitations of SalesBot compared to professional salespeople. The evaluation reveals gaps in recommendation quality and examines the challenges of ensuring content faithfulness in the CRS context. We emphasize the importance of considering the implications of generative AI systems in domains like conversational sales, where there several ethical concerns may arise related to upselling and providing factually accurate information.

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

## A  Product Catalog Generation

For each product domain, we automatically generate Product Catalog following the procedure in Table 4.

---

**1. Product Name Gen.**

Generate a list of top 30 [PRODUCT_NAME] options in the following format:

{"name": ...}

Include a diverse list of options with different brands, sizes, price points for a variety of customers.

---

**2. Product Metadata Gen.**

Generate product description, features, and price based on the product name. Output should be in the following json format:

```
{
 "name": "..."
 "price": "..."
 "description": "..."
 "features": [...]
}
```

Table 4: Product Catalog Generation Prompts.

An example of the resulting product metadata is presented in Table 5.

---

| | |
|---|---|
| **name** | Samsung 55" Class S95B OLED 4K Smart Tizen TV |
| **price** | $1,699.99 |
| **description** | Samsung OLED TV changes the game again with 8.3 million self lit pixels and ultra powerful 4K AI Neural Processing, all for a picture so real, it's surreal. Add on Dolby Atmos ® sound built in, the latest Smart TV apps, and a LaserSlim design and get a viewing experience that's intensely cinematic. |
| **features** | ["55 inch", "OLED Technology", "Neural Quantum Processor with 4K Upscaling", "Smart Calibration", "Connectivity with Bluetooth, RF, Wi-Fi, USB, HDMI, Ethernet (LAN), Digital Audio Out x 1 (Optical)", "Supported internet services: Netflix, Google TV, Amazon Instant Video, YouTube, Browser"] |

Table 5: Product Metadata Example for the TV category.

## B  ShopperBot Design

### B.1  Prompt

```
You are shopping online for a {product}. You haven't done your
   research on this product and want to speak to a salesperson over
   chat to learn more and make an informed decision.
Follow these rules:
- Chat with the salesperson to learn more about {product}. They will
   be acting as a product expert, helping you make an informed
   purchasing decision. They may ask you questions to narrow down
   your options and find a suitable product recommendation for you.
- Use your assigned preferences and incorporate them in your
   responses when appropriate, but do not reveal them to the
   salesperson right away or all at once. Only share a maximum of 1
   assigned preference with the salesperson at a time.
- Let the salesperson drive the conversation.
```

- Ask questions when appropriate. Be curious and try to learn more
  about {product} before making your decision.
- Be realistic and stay consistent in your responses.
- When the salesperson makes a recommendation, you'll see product
  details with 'ACCEPT' and 'REJECT' in the message. Please consider
  whether the product satisfies your assigned preferences.
- If the recommended product meets your needs, generate [ACCEPT]
  token in your response. For example, "[ACCEPT] Thanks, I'll take
  it!".
- If the recommended product is not a good fit, let the salesperson
  know (e.g. "this is too expensive")
- If you're not sure about the recommended product, ask follow-up
  questions (e.g. "could you explain the benefit of this feature?")
- Do not generate more than 1 response at a time.

Your assigned preferences:
{preferences}

Follow the above rules to generate a reply using your assigned
  preferences and the conversation history below:

Conversation history:
{chat_history}
Shopper:

## B.2 Architecture

Figure 4 highlights ShopperBot's key components and the overall design flow.

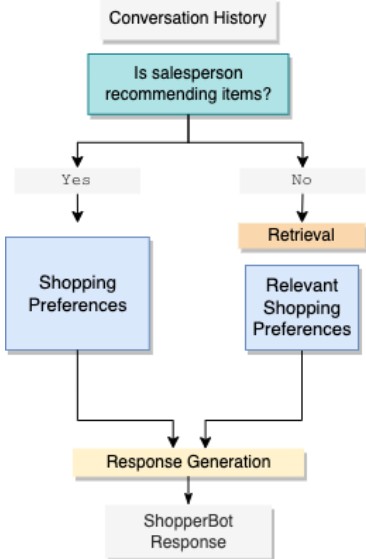

Figure 4: ShopperBot architecture to generate a response based on the selected shopping preferences.

## C   Human Evaluation Questionnaire

We recruit 150 crowd workers from Amazon MTurk to complete a survey. The user interface is presented in Figure 5. Examples of True/False quiz questions per product category are shown in Table 6.

## Survey

MTurk WorkerID:

X430YABCD

Answer the questions based on the chat transcript on the right ➡️

**[Communication]**

How would you rate salesperson's communication skills?

5 - Excellent ▾

Do you think the salesperson is:

Human ▾

Do you think the shopper is:

Human ▾

**[Informativeness]**

The resolution of a TV refers to the number of vertical pixels or dots that make up the TV's picture.

◯ Cannot answer based on the chat  🔘 True  ◯ False

A 4K TV with an OLED display technology and a refresh rate of 120Hz will generally have a better picture quality than a 1080p TV with an LCD display technology and a refresh rate of 60Hz

🔘 Cannot answer based on the chat  ◯ True  ◯ False

TVs with bigger screens always have better sound quality.

◯ Cannot answer based on the chat  ◯ True  🔘 False

**[Product Recommendation]**

Preferences:

```
▾ {
    "What is your budget?" : "around $2,000"
    "What size TV are you looking for?" :
    "a large size, maybe between 65 and 75 inches"
    "Is image quality a priority for you?" :
    "very important"
    "Do you have any preferred brands?" :
    "I like LG or Samsung"
    "Does your space have a lot of natural light?" : "yes"
}
```

## Sales Chat Transcript

Please read the chat below carefully, before answering questions.

Hi! How can I help you this afternoon?

Hi there! I'm actually shopping for a new TV and I'm not quite sure what I'm looking for. Can you tell me a bit more about the different types of TVs you have available?

Certainly! Choosing what size of TV to buy is based on your personal size preference, your room size, and your mounting options. You'll want to keep in mind the width and weight of the TV as that can affect whether you can mount it on your wall or you need an entertainment stand. Televisions can be as small as about 30 inches and as large as 80 inches (or more!). Do you have a size in mind?

I'm thinking something on the larger side, maybe between 65 and 75 inches. What options do you have in that size range?

I have TV's ranging in price from $749 to $9,999 in that size range. What will you be using the TV for mostly?

Well, I'm a bit of a picky shopper and I want to make sure I get something that will last a while. I watch a lot of movies and some sports, so I want something with good picture quality and a high refresh rate. Do you have any recommendations that might fit those preferences?

This 65 inch LG has a 120hz refresh rate that would be GREAT for watching movies and sports!

Product Name: LG - 65" Class NanoCell 99 Series LED 8K UHD Smart webOS TV
Price: $1,799.99
Description: Experience everything in more detail with LG NanoCell 8K. LG's most advanced 8K AI processor optimizes composition, black, light and color in every scene, while the

Figure 5: Human Evaluation User Interface.

| Product Category | Question | Answer |
|---|---|---|
| TV | The resolution of a TV refers to the number of vertical pixels or dots that make up the TV's picture. | T |
| | A 4K TV with an OLED display technology and a refresh rate of 120Hz will generally have a better picture quality than a 1080p TV with an LCD display technology and a refresh rate of 60Hz. | T |
| | TVs with bigger screens always have better sound quality. | F |
| Vacuum Cleaners | A canister vacuum is best suited for a large home with multiple floors and different types of flooring. | T |
| | If you have pets, it's important to choose a vacuum cleaner with a HEPA filter to capture pet hair and dander. | T |
| | Stick vacuums are not suitable for cleaning homes with stairs. | F |
| Mattress | Memory foam mattresses mold to the shape of your body. | T |
| | Gel memory foam mattresses are ideal for hot sleepers who want to stay cool at night. | T |
| | A mattress with more coils is always better than a mattress with fewer coils. | F |
| Guitar | The acoustic guitar is the most common guitar for first-time buyers. | T |
| | Acoustic guitars are self-amplified instruments that come in a variety of sizes, and the most common shape is the Parlor. | F |
| | Acoustic-electric guitars are the easiest way to perform with an acoustic guitar, and they always come with an integrated pickup system. | F |
| Laptop | A laptop with a larger screen size is always better for gaming. | F |
| | A laptop with a dedicated graphics card is unnecessary for basic office tasks. | T |
| | An SSD provides faster storage than an HDD. | T |
| Coffee Makers | Drip coffee makers are ideal for making coffee for one person. | F |
| | Single-serve coffee makers are convenient because they require no measurement of coffee grinds or water. | T |
| | Espresso machines can only make espresso shots. | F |

Table 6: True/False quiz questions per product category.

# D   SalesOps Framework User Interface

We present the SalesOps Framework's User Interface in Figure 6, displaying both the Seller View (a) and the Shopper view (b).

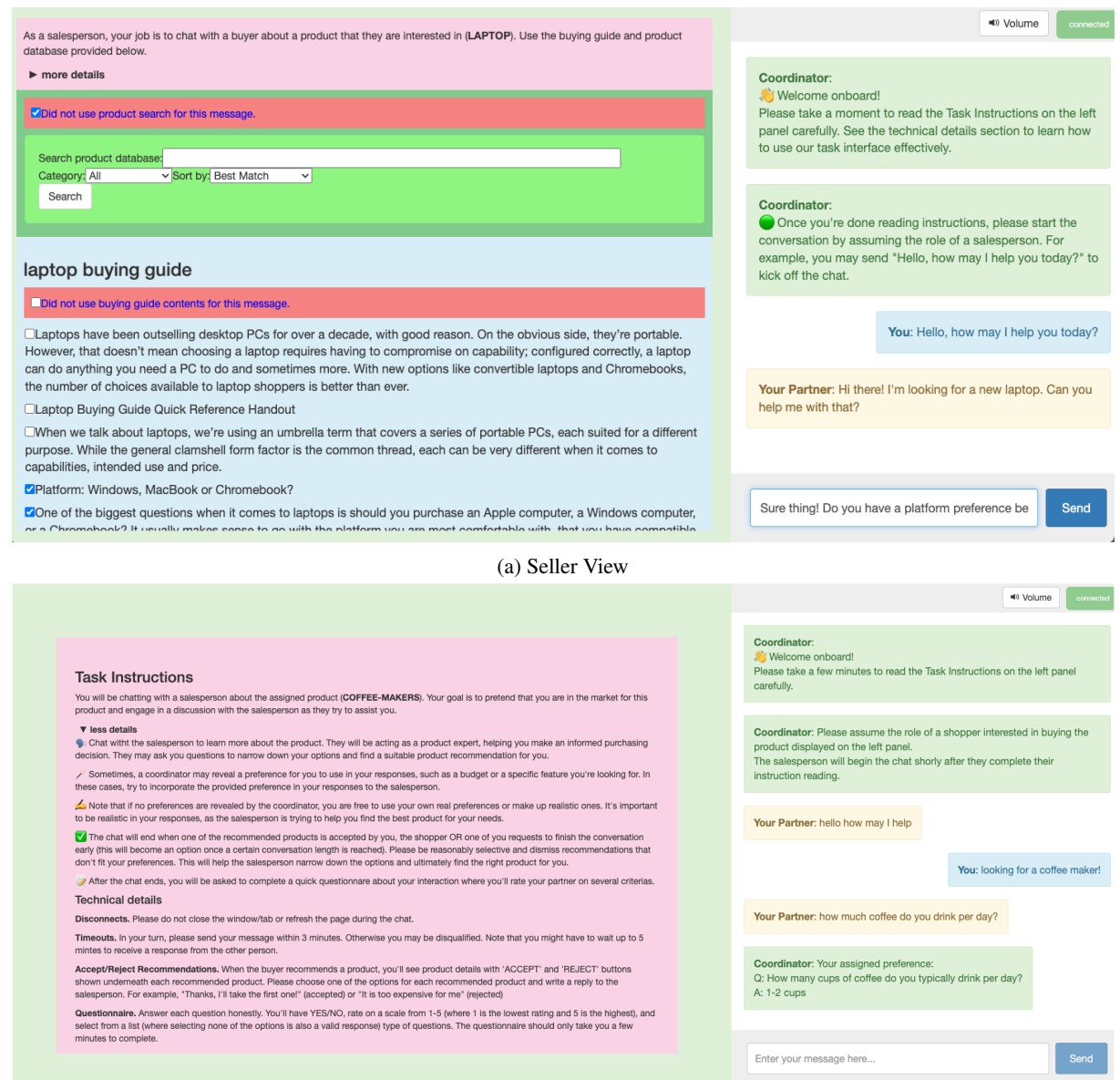

(a) Seller View

(b) Shopper View

Figure 6: SalesOps Framework User Interface.