# OpenReview forum: "Salespeople vs SalesBot: Exploring the Role of Educational Value in Conversational Recommender Systems"
_EMNLP/2023/Conference — EMNLP 2023 Findings_

### Official Review · Reviewer_Uvkj · 2023-07-31

**Soundness:** 5

**Excitement:**

4: Strong: This paper deepens the understanding of some phenomenon or lowers the barriers to an existing research direction.

**Missing References:**

The evaluation criteria need more background and some more reasoning why the are useful to evaluate the interactions.

Starting points could be the ITU-T Recommendation P.852 : Subjective quality evaluation of text-based chatbots (https://www.itu.int/rec/T-REC-P.852-202207-I) and Supplement 24 to ITU-T P series P.Sup24 : Parameters describing the interaction with spoken dialogue systems (https://www.itu.int/rec/T-REC-P.Sup24-200510-I)


**Paper Topic And Main Contributions:**

The author describe the setup and analysis of an NLP engineering experiment. I general, the paper describes the usage of a LLMs to generate both interlocutors for a dialog system (here a chatbot) and to use a LLM (large language model) for the evaluation of generated dialogs.

More specific, the authors have created a chatbot (SalesBot) for the sales domain and a user simulation (Shopper) which can interact with the each other. For both, the system and the user simulation, LLMs are used in an instructive way, using prompting only. A special goal, in the shopping domain, is to educate user about (type of) products while asking for recommendations.

To evaluate the generated interactions, evaluation criteria and related metrics have been developed. The metrics can be used by humane evaluators as well as again a LLM.

For evaluation 150 dialogs have been generated and rated by a LLM - the recommendation quality was rated by the authors. Furthermore, additional 150 interactions have been generated using 15 human salesman (replacing SalesBot) and  Shopper.

**Questions For The Authors:**

Question A: What are the Inf_q questions?
Question B: What is the number of annotators for Rec. quality (*Rec*)?
Question C : What is the annotator agreement for *Rec*?


**Reasons To Accept:**

- The authors demonstrate that a completely LLM-based development and evaluation of chatbots for the sales domain is possible and leads to good results
- The authors present a well planned evaluation of SalesBot and Shopper which incorporates a comparison of the the setup with human salesman
- The paper gives important stimuli for the generation and evaluation of chatbots using LLMs in a zero-shot learning manner in other domains

**Reasons To Reject:**

- The evaluation covers all important general aspects but is not described in all necessary details (the quiz questions are not presented)
- A motivation and derivation of the evaluation criteria and metrics is missing
- The title of the paper puts the focus on educational value, which is finally only one (side) aspect among many other topics in the paper
- It is not evaluated if the metric measures the relevant factors to evaluate interactions in the sales domain


**Reproducibility:**

4: Could mostly reproduce the results, but there may be some variation because of sample variance or minor variations in their interpretation of the protocol or method.

**Reviewer Confidence:**

5: Positive that my evaluation is correct. I read the paper very carefully and I am very familiar with related work.

**Typos Grammar Style And Presentation Improvements:**

- The sentence starting in line 661 needs a verb.

---

> ### Author Rebuttal · Authors · 2023-08-28
>
> Thank you for the insightful comments, questions, and feedback on this work!
>
> Regarding human evaluation details (Questions A/B/C), these are all great points. We will modify the Appendix to include all of the Inf_q
>  quiz questions (e.g., "Acoustic guitar is the most common guitar for first-time buyers. [TRUE] or [FALSE]") and add the number of MTurk workers who completed the survey (150).
>
> Regarding the evaluation criteria, in Section 5 we briefly provide the motivation behind our metrics of: (1) informativeness - driven by our educational objective, (2) fluency - standard for any generative system, and (3) recommendation quality - standard for any recommendation system. We will also add Future Work subsection under Discussions which will include user satisfaction as an additional evaluation criteria to enable more iterative improvements of CRS.

---

### Official Review · Reviewer_LBGM · 2023-08-01

**Soundness:** 3

**Excitement:**

3: Ambivalent: It has merits (e.g., it reports state-of-the-art results, the idea is nice), but there are key weaknesses (e.g., it describes incremental work), and it can significantly benefit from another round of revision. However, I won't object to accepting it if my co-reviewers champion it.

**Paper Topic And Main Contributions:**

In this paper, the authors have experimented with utilising LLM and Information Retrieval system to create a SalesBot, a chatbot that could help a customer choosing the right product to buy, and a ShopperBot, an LLM module that learns about the customer's preference based on the customer's answer and the SalesBot information.
This paper is a position papers on NLP engineering experiment.

**Questions For The Authors:**

1. Have you compared your system with other sales chatbots or recommender systems, including the ones that you have mentioned in your Related Work sections?
2. In Table 1, it seems that using rule-based increase the informativeness of your system, why didn't you use this as the final configuration?
3. What retrieval system did you use for both the Seller and the Shopper?
4. Have you done ablation study on your Shopper Bot?
5. What do you think we should do to improve the Seller Bot and Shopper Bot performace?

**Reasons To Accept:**

* The author has done a thorough literature review and a thorough evaluations on their ideas.
* The idea of using a ShopperBot and a SellerBot to help a customer in eCommerce situation is interesting.

**Reasons To Reject:**

* Although the authors' proposed framework, as expected, provides a fluent chat responses, the quality of it's recommendation and informativeness are significantly below human salesperson
* The proposed framework is not that novel
* The authors have not compared their framework with other chatbots or recommender systems
* The authors have not done a thorough discussions on what need to be done to improve the performance of a salesbot in this area

**Reproducibility:**

3: Could reproduce the results with some difficulty. The settings of parameters are underspecified or subjectively determined; the training/evaluation data are not widely available.

**Reviewer Confidence:**

4: Quite sure. I tried to check the important points carefully. It's unlikely, though conceivable, that I missed something that should affect my ratings.

---

> ### Author Rebuttal · Authors · 2023-08-28
>
> Thank you for taking the time to review our paper and provide actionable feedback!
>
> Regarding the fact that the SalesBot system does not achieve on-par performance with expert human sellers. This is indeed a major finding of our work and showing that AI can outperform human sales experts is not the goal of this work. Instead, we hope that the identified gap and our opensource framework can help direct future research towards the most important problems in this space. We will address this point in our Discussion section.
>
> We believe that our novelty comes from the SalesOps framework, the overall task definition, and the findings from our user study. To the best of our knowledge, autonomous LLM-based agents like SalesBot have not yet been applied to CRS. Additionally, our SalesOps framework goes beyond prior work in terms of setup (with the gradual preference revelations and several knowledge sources provided) and the multi-aspect evaluation. We will improve the Related Work section to clearly reflect the novelty in our framework and individual system implementations.
>
> Q1: Regarding comparison to other chatbots/recommender systems, we agree that we could add more detailed information on the related work. In our preliminary experiments, we included a baseline which relied on BlenderBot3 3B (Shuster et al., 2022), however due to its model’s limited context length, we found it underperformed text-ada-001. Only recent models have the capacity to handle large context length windows required in our framework (due to Product DB), which limits our ability to compare to older systems. The ablation study we conducted includes common baselines as replacements to individual components of SalesBot, which provide insights into the necessity of the LLM in the components of SalesBot.
>
> Q2: We found that the rule-based action decision module leads to improved informativeness (as the system relies more heavily on the knowledge action), but at the cost of fluency and recommendation quality. We argue that recommendation quality should not be compromised to provide educational value, so we steer away from hard coding a set of knowledge-based turns into a system. This is also shown in a decrease in fluency, meaning that users of the system would quickly recognize these rules which might lead to unsatisfactory outcomes. Diving deeper into user satisfaction would be a promising direction of future research for CRS evaluations.
>
> Q3: We implemented retrieval using FAISS and "sentence-transformers/all-mpnet-base-v2" embedding model (sections 4.2.2 (knowledge search) and 4.2.3 (product search)).
>
> Q4: No, we mainly focus on SalesBot performance and leave the ShopperBot optimization to future work in User Simulation research.
>
> Q5: We will add a subsection to our Discussion section dedicated to Future Work. This section will propose 3 directions that we believe would lead to further improvements in systems similar to SalesBot. In summary, they are (1) applying faithfulness methods to CRS, (2) optimizing retrieval for recommendation quality, and (3) measuring user satisfaction to enable more iterative improvements of CRS.

---

### Official Review · Reviewer_fUC4 · 2023-08-02

**Soundness:** 2

**Excitement:**

3: Ambivalent: It has merits (e.g., it reports state-of-the-art results, the idea is nice), but there are key weaknesses (e.g., it describes incremental work), and it can significantly benefit from another round of revision. However, I won't object to accepting it if my co-reviewers champion it.

**Paper Topic And Main Contributions:**

The authors propose a conversational recommender systems (CRS) providing both product recommendations and educational value through mixed-type mixed-initiative dialog in e-commerce domain. By analyzing the necessary components of CRS, the authors utilize the LLM technology to build SalesBot and ShopperBot to simulate both sides in product purchasing. Experimental results reveal that SalesBot approaches professional performance in terms of fluency and informativeness, but lags behind in recommendation quality.

**Reasons To Accept:**

- The authors present a novel idea of exploring the role of educational value in CRS, and the idea of utilizing LLMs to build this CRS seems technically sound.
- The authors use a combination of human evaluation and NLI-based analysis to compare SalesBot and professional salespeople, providing a comprehensive evaluation of the agents.
- The authors provide insights of the limitations of both SalesBot and professional salespeople in providing truthful information and how these challenges can be addressed in the CRS context.


**Reasons To Reject:**

- The content organization can be better improved to increase the readability. More details on the SalesBot and ShopperBot design and development should be given to increase the clarity. A technical figure describing the proposed framework is missing, and thus fails to better elaborate how different component could cooperate with each other.
- The sample size of products and professional salespeople used in the work is relatively small, which may limit the generalizability of the results.
- The result of the ablation study can be better analyzed, and the potential usage of the findings for conversational agent design should be clearly addressed.


**Reproducibility:**

4: Could mostly reproduce the results, but there may be some variation because of sample variance or minor variations in their interpretation of the protocol or method.

**Reviewer Confidence:**

3: Pretty sure, but there's a chance I missed something. Although I have a good feel for this area in general, I did not carefully check the paper's details, e.g., the math, experimental design, or novelty.

---

> ### Author Rebuttal · Authors · 2023-08-28
>
> Thank you for a thorough review of our work! We have carefully reviewed your feedback and would like to address the following points.
>
> Regarding the readability, we will modify and rethink the presentation of Figure 1 which depicts the overall framework. Figure 3 currently provides the technical components of SalesBot, and we will add a similar Figure for the ShopperBot system in the final version.
>
> Regarding sample sizes: we have 6 product categories (e.g., vacuum, TV, etc) each containing 30 products (e.g., iRobot Model X, etc), so in total 180 products. Limiting the number of products allowed salespeople to familiarize themselves with the product offering prior to selling.
> We recruited 15 professional salespeople to conduct experiments. We preferred to recruit fewer people and train them well in order to obtain high quality chats. With our current number of participants, we observe meaningful quantitative differences between professionals and SalesBot. The goal of our work is not to collect a large-scale dataset or conduct a large-scale user study, instead we focus on the overall task definition, SalesOps framework design, and a first glance at sales automation.
>
> We agree with the reviewer that the presentation of our Ablation results (Section 6.2) can be improved by providing concrete recommendations for designers of conversational agents. Namely, we will add the following recommendations: (1) Leverage LLMs in Response Generation (2) Balance Informativeness and Fluency through Action Decision Module (3) Integrate Generative Flexibility in Retrieval Queries (4) Utilize Regeneration for Improved Fluency.

---

### Meta-Review · Area_Chair_2HoC · 2023-09-18

**Recommendation:** 3

**Metareview:**

This work utilizes large language models (LLMs) to construct a SalesBot and ShopperBot to simulate both sides in product purchasing. One aim of the work is to introduce a new paradigm for conversational recommender systems (CRS) which, in addition to providing product recommendations, includes customer education.

This work makes several contributions, including:
 - Presentation of a novel idea of exploring the role of educational value in CRS
 - Demonstrates utilization of LLMs to build CRS
 - Conducts a study that compares SalesBot against professional salespeople

The result are however largely in line with expectations for LLMs, i.e, while it provides fluent chat responses, the quality of it's recommendations and informativeness are significantly below those of a human salesperson. Additionally, there is a lack of comparisons with other relevant chatbot/CRS systems and analysis of the ablation studies to better inform conversational agent design.

---

### Meta-Review · Senior_Area_Chairs · 2023-10-05

**Recommendation:** 3

**Metareview:**

meta review

---

### Decision · Program_Chairs · 2023-10-07

**Decision:**

Accept-Findings

**Comment:**

This work utilizes large language models (LLMs) to construct a SalesBot and ShopperBot to simulate both sides in product purchasing. One aim of the work is to introduce a new paradigm for conversational recommender systems (CRS) which, in addition to providing product recommendations, includes customer education.

This work makes several contributions, including:
 - Presentation of a novel idea of exploring the role of educational value in CRS
 - Demonstrates utilization of LLMs to build CRS
 - Conducts a study that compares SalesBot against professional salespeople

The result are however largely in line with expectations for LLMs, i.e, while it provides fluent chat responses, the quality of it's recommendations and informativeness are significantly below those of a human salesperson. Additionally, there is a lack of comparisons with other relevant chatbot/CRS systems and analysis of the ablation studies to better inform conversational agent design.|meta review